# Challenges Regarding Transition from Case-Based Learning to Problem-Based Learning: A Qualitative Study with Student Nurses

**Ramoipei J. Phage \***[ID]**, Boitumelo J. Molato and Molekodi J. Matsipane**

NuMIQ Research Focus Area, School of Nursing, Faculty of Health Sciences, North-West University, Ngaka Modiri Molema 2735, South Africa
* Correspondence: msg.paghe@gmail.com

**Abstract:** Background: The transition from case-based learning to problem-based learning can be challenging and may have negative effects on the academic, psychological, emotional, or social well-being of student nurses. As a result, this exposes student nurses to high failure rates, anxiety disorders, a loss of uniqueness, and fear of the unknown. However, student nurses employ different strategies aimed at overcoming challenges faced during this transition period. Methods: An exploratory, and descriptive research approach was used. A purposive non-probability sampling technique was used to select participants. Focus group discussions via Zoom video communication were used to collect data, which were analysed using Braun and Clarke's six steps of thematic analysis. Results: The following three themes emerged: challenges regarding facilitation, challenges regarding assessment, and strategies to overcome challenges. Conclusions: The study established that student nurses are faced with different challenges during the transition from one teaching strategy to another. Student nurses suggested strategies that could be used to overcome these challenges. However, these strategies are not enough and therefore more needs to be done to support and empower student nurses.

**Keywords:** case-based learning; challenges; problem-based learning; student nurses; transition

## 1. Introduction

Various active learning methods are used in the teaching and learning process in order to produce students who are creative, adaptive to team work and are able to find solutions to the problems of daily life by using the knowledge and skills gained [1]. Case-based learning (CBL) and problem-based learning (PBL) are among these methods of teaching and learning [1]. Keeping in view all the aforementioned facts, Liu et al (2019) asserted that CBL and PBL are long established pedagogic teaching strategies that have been widely implemented throughout various higher institutions of learning globally [2]. In a study conducted in China by Bi et al., (2019), CBL is an active teaching and learning strategy that focuses on student nurses as the centre of the learning environment [3].

Moreover, CBL encourages a student-centred and patient-oriented exploration of realistic and specific situations [3]. The authors also mention that student nurses focus on the patient's case, engage in self-directed learning, scientific inquiry, and collaboration with others in integrating theory and practice [1,3]. Furthermore, Hassoulas et al. (2017) state that CBL provides a practical model for students to relate content learning to professional practice and helps them improve the ability to collaborate when studying, thinking critically, and solving clinical problems [4]. Despite these benefits of CBL, there are also related challenges that student nurses may encounter which include: students feeling that CBL activities in the classroom take a lot of time, and other students finding it uncomfortable to engage in group learning activities since they prefer working independently [5].

Historically, CBL was invented in the United States of America (USA) in 1870 at Harvard University Law School, more than a century before PBL [6]. As a result, this endorses the fact that institutions of higher learning have indeed been using these two teaching methods for a very long time. According to literature, PBL was used initially in nursing education in the year 1969 at McMaster University Medical School in Canada [6]. PBL is a student-centred pedagogical approach in which students learn a subject through the experience of solving an open-ended problem found in trigger material [7]. PBL is known to increases student motivation and desire to learn, and strengthens cooperative learning skills [8]. Therefore, students are more likely to become active in their learning. Despite all these advantages, PBL has inherent limitations and disadvantages that cannot be overlooked. For example, PBL requires a significant amount of time, there is lack of training on PBL facilitation, and it is expensive to implement, to mention but a few [8].

In the South African (SA) context, Rakhudu et al. (2016) reported that five higher institutions of learning have adopted PBL as their teaching strategy [9]. Amongst five institutions that have adopted PBL in South Africa, there is one institution of higher learning in the North West Province (NWP) that has adopted both CBL and PBL as its teaching strategies in the undergraduate programme since 2002. This particular institution applies CBL in the first two years of study. During the third and fourth years, the same student nurses are expected to transition from CBL to PBL, which may cause anxiety to the student nurses due to a fear of anticipated challenges that may accompany a transition from one teaching strategy to another.

Since the implementation of both CBL and PBL in 2002 at a particular institution of higher learning in the NWP, South Africa, there has been no research study that has been conducted to explore and describe the challenges that student nurses may encounter during the transition period, thus resulting in a dearth of literature regarding their challenges. Therefore, the researchers deemed it necessary to explore and describe challenges faced by student nurses who transition from CBL to PBL in order to have a deep understanding from the student nurses' perspective with the aim of improving the teaching and learning process.

Transition is described as an internal, psychological process that results from a change. The actual change may occur quickly, but the transition process occurs much more slowly and is different for everyone [10]. The study conducted by Jindal-Snape et al. (2019) acknowledges that transition may have negative impact on the well-being of students, either psychological, emotional, or social [11]. Moreover, a transition from one teaching strategy to another exposes students to increased failure rates, anxiety disorder, a loss of uniqueness, and fear of the unknown [10]. The researchers are of the view that there might be challenges if the student nurses are to transition from the CBL to PBL teaching strategy. The researchers garner this view from the understanding that CBL and PBL as teaching strategies have their own challenges that may be compounded by the process of transitioning which on its own is a challenge to most students. Furthermore, the researchers believe that a profound understanding about the challenges faced by student nurses during transition from CBL to PBL in the NWP of South Africa is necessary in a sense that if they are successfully explored, this may assist in finding efficient and effective solutions to mitigate the challenges.

## 2. Material and Methods

### 2.1. Aim

The aim of this study was to explore and describe challenges faced by student nurses during transition from case-based learning (CBL) to problem-based learning (PBL) at one of the institutions of higher learning in the North West Province (NWP).

### 2.2. Research Design

An exploratory, and descriptive qualitative research approach was used to explore, describe, and contextualize challenges faced by student nurses during transition from CBL to PBL at one of the institutions of higher learning in the NWP, South Africa. An exploratory,

descriptive, and contextual research approach was conducted to gain a new insight, and discover new ideas, thus increasing the knowledge about the phenomenon at hand.

### 2.3. Study Setting

The study was conducted at one of the institutions of higher learning in the NWP, South Africa, at which case-based learning and problem-based learning are used to facilitate teaching and learning. The study was conducted only in the Ngaka Modiri Molema (NMM) district in the NWP, because that is where the institution is situated. This institution has a capacity to accommodate four hundred student nurses on average. The higher institution is presently offering the undergraduate program Bachelor of Nursing Science (BNSc), whereby English is used as a medium of instruction. The BNSc program takes a minimum of four years to complete, from the first to fourth year level. Currently, the institution of higher learning has a total number of three hundred and nine student nurses.

### 2.4. Population and Sampling

The population of this study included student nurses in their third year of study who used CBL and are currently using PBL as their teaching and learning strategy. Although both the third and fourth-year student nurses have been exposed to CBL and PBL, the third-year student nurses were the only population of this study because they were recently exposed to PBL, which is new to them. As a result, the researcher believed that the third-year students were in a more vulnerable position than the fourth years as far as the transition from CBL to PBL was concerned. Furthermore, the researchers were of the opinion that the fourth-year students might have already grown used to PBL; hence, they were excluded from the population. A purposive non-probability sampling technique was used to sample the participants. Informed consent forms with additional important information about the study was emailed to the participants to read and sign. Each participant brought the signed consent form on the day of focus group. The sample size of the study was determined by data saturation.

### 2.5. Data Collection Method

Data were collected using semi-structured focus group discussions. Since there was the coronavirus (COVID-19) pandemic and mass gathering was prohibited, the researchers conducted the focus groups via Zoom video communication to avoid physical contact with participants, with the aim of minimizing the possible exposure to COVID-19.

Open-ended questions were asked and follow-up questions using probing, clarifying, and other communication techniques were used to improve communication dynamics. Based on Krishnasamy et al (2023), an example of a focus group questions is shown below in the form of Table 1 [12].

**Table 1.** Example of focus group questions.

| Exploratory Questions |
| --- |
| • What are the challenges you faced during transition from Case-based to Problem-based learning? <br> • What helps you to overcome the challenges? <br> • What else do you think other people can do to overcome the challenges? |

Focus group sessions were audio-recorded with the permission of participants and transcribed verbatim into transcription data. The transcribed data were de-identified to maintain a principle of anonymity. Furthermore, all data was kept in a lockable safe place to promote privacy and confidentiality.

## 2.6. Data Analysis

The researchers and co-coder analysed data independently using Braun and Clarke's six steps of thematic analysis. The researchers prepared and organised data from all Zoom video communications. The researchers transcribed data from all Zooms and sent the transcribed data to the independent co-coder for coding. Data was coded using inductive process. Themes and subthemes were generated and can be viewed in Table 2 under results. Both the researchers and the co-coder held a meeting via Zoom video communication to discuss the themes and subthemes and agreed upon the themes and subthemes. Subsequently, the researchers narrated those themes and subthemes, quoted the participants, and supported them with the literature under discussions.

**Table 2.** Themes and subthemes.

| Theme | Sub-Themes |
|---|---|
| 1.1 Challenges regarding facilitation | 1.1.1. Adaptation challenge<br>1.1.2. Group work<br>1.1.3. Information search<br>1.1.4. Workload and insufficient time for PBL content<br>1.1.5. Lack of proper guidance from lecturers<br>1.1.6. Learning issues<br>1.1.7. Online problem-based learning<br>1.1.8. Lack of student instructors (SI) |
| 1.2 Challenges regarding assessment | 1.2.1 Lack of revision<br>1.2.2. Lack of feedback |
| 1.3 Strategies to overcome challenges | 1.3.1 Collaborating with classmates/Peer learning<br>1.3.2. Use of relevant articles, prescribed books, and previous question papers<br>1.3.3. Plan study time<br>1.3.4. Consultation with lecturers |

## 2.7. Ethical Considerations

The researchers submitted the research proposal to the North West University's scientific committee, Research Data Gatekeeper Committee (RDGC), and Health Research Ethics Committee (Ethics reference number: NWU-00218-21-s1), respectively, for ethical approval. Before the recruitment of participants and data collection, the study received unanimous approval from all the aforementioned committees. Participants were recruited via a wide range of methods, and all their rights were explained to them before signing consent forms. Participants signed consent forms before data collection and were addressed by alphabet rather than their names to maintain anonymity throughout the study. Furthermore, participants were also informed that they were free to withdraw from the study without consequences.

## 3. Results

A total of six focus group interviews consisting of six to eight participants via Zoom communication video were conducted and data saturation was reached with the sixth focus group. The results of the research study are discussed in to Table 2, which provides themes and subthemes. There are three themes and thirteen subthemes that emerged from the results of the study.

Student nurses expressed their challenges regarding the transition from case-based learning to problem-based learning in different ways. These challenges include facilitation challenges, assessment challenges, and strategies to overcome challenges. Each form of challenge faced by student nurses as a result of the transition were grouped into subthemes and discussed independently as follows:

*3.1. Theme 1.1: Challenges Regarding Facilitation*

The first theme focuses on the challenges regarding facilitation. This theme consists of seven sub-themes and are as follows: adaptation challenges, group work, information search, increased workload and insufficient time for problem-based learning content, lack of guidance from lecturers, learning issues, and online problem-based learning.

### 3.1.1. Subtheme 1.1.1: Adaptation Challenges

Adaptation challenges regarding the transition from CBL to PBL was identified as one of the subthemes. Participants revealed that it is challenging to adapt from one teaching strategy to the other. For instance, learning with PBL as a teaching strategy in third year while being used to a CBL teaching strategy from the first to second year of study. Participants expressed their views as follows:

*"The challenge that I faced during transition was to adapt from case-based to problem-based learning."* (**Participant U**).

Another participant supported the statement made by participant U and added:

*"Well, my point supports the statement of student U. The main problem is that we struggled to adapt."* (**Participant Y**).

Another participant described why they had a challenge regarding adaptation and said:

*"With case-based learning, we were face to face with the lecturers. Lecturers guided us that is why we adapted very easily into case-based learning unlike in problem based learning."* (**Participant W**).

### 3.1.2. Subtheme 1.1.2: Group Work

Working in groups emerged as a challenge student nurses faced the during transition period. Additionally, student nurses added that the poor participation of some of the group members, and changing of group members on a yearly basis are some of the reasons leading to group work challenges. Participants talked about the issue of group work versus poor participation and articulated the following:

*"When we transit to problem-based learning, we were given problems to solve and we were working in groups. Some of the students did not want to participate, end up having to go to class unprepared, not being able to solve the given problems. So, the biggest problem was working in groups."* (**Participant W**).

Another participant concurred with the statement made by participant W and said:

*"The only issue that we had is poor participation from some group members. You gather your own information and end up doing the other member's part of the job delegated to him/her."* (**Participant D**).

One of the participants explained how the issue of changing group members impact on group related work and added:

*"The groups are changing every year. In the beginning of the year, we expect to have challenges because we are not used to each other as we are all new in the group and it is a big challenge. But as time goes on it becomes better since we get used to each other as group members. We get to know strengths and weaknesses of each other and we are delegating task based on strength and weaknesses. It becomes much easier to work together in that way. But now next year is the same routine again where I will get a new group and have to start the process all over again."* (**Participant A1**).

### 3.1.3. Subthemes 1.1.3: Information Search

Searching for and finding relevant information on databases or search engines was another aspect that emerged as a subtheme. On top of that, student nurses added that there is lack of judgment from the lecturers as to whether the acquired information is relevant or not. Below are some of the views the participants mentioned:

*"You have to go and look for information and sometimes you don't even know the sites (database) to get information from. You find out that sometimes you didn't get enough information and feedback from the lecturer and you don't know if you are right or wrong."* (**Participant N**).

Another participant described looking for information as a challenge and said:

*"Problem-based learning was a whole new experience on its own as we look for information ourselves which is a challenge."* (**Participant A5**).

### 3.1.4. Subtheme 1.1.4: Workload and Insufficient Time for PBL Content

Student nurses reported that an increased workload accompanied by insufficient time for PBL content was a challenge aroused during transition from CBL to PBL. Most of the participants indicated that:

*"It (PBL) needs you to be very flexible of which it comes with a lot of work and we don't have much of time because we are doing practicals as well as theory."* (**Participant K**).

The views shared by other participants were:

*"With PBL we do a lot of different topics in a short period of time and we end up struggling remembering all the condition, we lose concentration."* (**Participant Q**).

*"Another challenge is that the PBL workload is too much, as a student the work load is always a challenge but in this case it was too much to a point that we ended up doing the work for the sake of just submitting."* (**Participant S**).

*"I don't know if it comes with problem-based learning but I feel like our lecturers gives us less time to do the work. They will give us work on Friday and say we must submit on Tuesday."* (**Participant H**).

### 3.1.5. Subtheme 1.1.5: Lack of Proper Guidance from Lecturers

A lack of proper guidance from lecturers emerged as one of the subthemes. Students raised concerns to say, since the transition from CBL to PBL, they are often not guided by lecturers when given homework. As a result, this poses a challenge. Some of the comments made by participants are as follows:

*"Most of the work is being done by the student without the guidance of the lecturers, it's more like you are your own lecturer."* (**Participant P**).

*"Sometimes we do not get clarity in class, we are usually given a topic or condition and we go and prepare when we come back to present, there is usually no clarity as to say what you did there was wrong and what you did there was right, you should go fix here and there."* (**Participant A2**).

*"Personally the challenges that I face with transiting to problem-based learning (PBL) is that, we are given a problem and we have to go and find solutions ourselves, then we present it to the lecturers after that we are not being corrected or told that this and this is wrong, they are a learning issue if we ask questions."* (**Participant Q**).

### 3.1.6. Subtheme 1.1.6: Learning Issues

Learning issues emerged as one of the subtheme. A "learning issue" in this context is homework given to student nurses in class if students do not reach a consensus about certain information presented. The majority of the participants raised the concern about the challenge regarding learning issues and commented as follows:

*"In problem-based learning, we facilitate everything ourselves. So we end up having learning issues, then we go research about them then come back the next day and present the same thing. We are not facilitated properly or corrected by the module facilitator in most cases actually hence we end up having learning issues."* (**Participant U**).

Other participants also commented on the problem of learning issues and said:

*"Regarding transition to problem-based learning, during our classes there can be something new that is raised and if we do not the answer in class, it becomes a learning issue for a very long time."* (**Participant A3**).

*"If we can't find the correct information or answer for that learning issue it becomes a learning issue for about three weeks."* (**Participant H**).

*"It can become a learning issue forever because none of us comes up with a direct answer or if we come up with the wrong answer during presentations then it's going to be a learning issue until we come up with the right answer."* (**Participant A6**).

Subsequently, another participant added and commented to say:

*"Adding on the learning issues, yes my colleagues are correct with this process of us getting learning issues every week instead of the lecturers correcting us-It is a problem because if we were supposed to finish one module in 10 weeks we end up taking longer to finish the module because of those learning issues."* (**Participant A1**).

### 3.1.7. Subtheme 1.1.7: Online Problem-Based Learning

Online problem-based learning facilitation was raised as one of the subthemes. Student nurses expressed their challenges relating to the conveyance of PBL via online platforms. As a result, this affected their transition to PBL. The majority of the participants raised a serious concern about PBL that was conducted online. The majority of participants have articulated that:

*"I feel like it is the online learning that is challenging so I think since we are going back to contact learning I think it is going to be much better."* (**Participant P**).

*"The online learning also contributed in us not really understanding the problem-based learning. Even now, we don't fully understand problem-based learning, because we have been doing it online."* (**Participant W**).

*"Sometimes you will find that we attend class while we are in our beds and you will fall asleep and when you wake up you did not hear anything that's the problem. Because we are attending online in our own space, we will be sleeping and the lecturers are not even aware of that."* (**Participant J**).

Other participants also raised their views and mentioned that:

*"We are using online platforms such as google meetings and some people have connectivity problems. They will be having the correct answer but due to connectivity problem, they cannot give that answer then it becomes a learning issue."* (**Participant A1**).

*"Remember most of the lessons are conducted online and we have network issues, so sometimes when there is an assistance we experience connectivity issues and end up missing that segment of the lesson."* (**Participant A5**).

*"Most classes are online and there are connectivity issue sometimes. This makes it hard to voice out your opinion, so we can't express ourselves like in a contact classes."* (**Participant P**).

One participant highlighted that she did not have a problem with the transition from CBL to PBL. However, the main challenge with PBL was that it was conducted online and they said:

*"I did not have problems or challenges with transition from case-based learning to problem-based learning, so for me the issue was more of the online problem-based learning."* (**Participant S**).

### 3.1.8. Subtheme 1.1.8: Lack of Student Instructors (SI)

A lack of student instructors (SI) also emerged as one of the subthemes. Student nurses expressed their concern relating to a lack of student instructors in their third year of

study, where they need them the most. In this context, a student instructor (SI) is any senior student that will help junior students with their studies by facilitating extra classes during their spare time. Students mentioned that in their first and second year, when they were using case-based learning, they used to have student instructors and that helped them a lot. For this reason, they believe if the same principle can be applied in third year, in which they use problem-based learning, it can help them adapt well to the PBL teaching strategy. Participants expressed their own view regarding the lack of student instructors (SI) and said:

*"it is not advise perse to the students but it's a suggestion on what the lecturers maybe can do or what the school of nursing can implement, like for our ancillary modules we used to have SI. So maybe if they can introduce SI's for third and fourth years whereby other students can actually conduct classes to explain further for those who needs further explanations. Like they are senior students so they are able to explain better to the junior students."* (**Participant G**).

Another participant added:

*"Sometimes I just wish like we had SI for certain modules, like other student who are doing different courses have SI to help them. So in this course we don't have SI who can help us with studying like other programs in the university."* (**Participant A4**).

*3.2. Theme 1.2: Challenges Regarding Assessment*

The second theme focused of the challenges regarding the assessment of student nurses and produced two sub-themes, namely a lack of feedback and revision. The sub-themes were discussed as follows:

3.2.1. Subtheme 1.2.1: Lack of Feedback

A lack of feedback from lecturers is another subtheme that emerged. Student nurses raised their concern to say in PBL, lecturers usually do not give feedback or correct them when they are wrong, or when they do, it is not sufficient. As a result, this impacts their assessment negatively. One participant uttered:

*"With lecturers in PBL we do not get correction or feedback."* (**Participant Q**).

Another participant added to say:

*"Sometimes you didn't get enough feedback from the lecturer and you don't know if you are right or wrong."* (**Participant N**).

3.2.2. Subtheme 1.2.2: Lack of Revision

A lack of revision was raised as one of the subthemes. Student nurses reported that there is sometimes less time to revise, which somehow results in a challenge during assessments. One participant mentioned that:

*"When we are busy with the learning issues, we can't move to other topics, that way we have less time to finish that module for that semester. In that way, we don't have enough revision time for exam."* (**Participant H**).

Another participants added:

*"Adding on the learning issues, yes my colleagues are correct with this process of us getting learning issues every week instead of the lecturers correcting us. It is a problem because if we were supposed to finish one module in 10 weeks we end up taking longer to finish the module because of those learning issues. The duration of a module becomes longer, it can take up to 16 weeks or 17 weeks and we end up finishing late and it waste the time we should be using revising the content of that module in preparation to the upcoming tests or exams."* (**Participant A1**).

*3.3. Theme 1.3: Strategies to Overcome Challenges*

The third theme focuses on the strategies used by student nurses to overcome challenges faced during the transition from CBL to PBL. This theme consists of four sub-themes, namely collaboration with classmates/peer-assisted learning, the use of relevant study materials such as articles, prescribed books and past question papers, the planning of study time, and consultation with lecturers. The subthemes are discussed as follows:

3.3.1. Subtheme 1.3.1: Collaboration with Classmates/Peer-Assisted Learning

Collaboration with classmates, also referred to as peer-assisted learning, emerged as one of the subthemes. Students expressed the use of peer-assisted learning as one of the strategies they used to overcome the challenges faced during the transition from case-based to problem-based learning. Different participants explained:

*"I will also ask the presentations of different groups and compile and study with it together with my own work because I can not only rely on my work to help me so that's how I manage to cope."* (**Participant N**).

*"Studying as a group also helps because discussing amongst ourselves also makes it easier."*
(**Participant Q**).

*"We as student decided to share the presentations and slides that we have with each other and we would discuss with each other or go into other platform like YouTube to watch videos that would give more information. It made it much better."* (**Participant S**).

Further participants added:

*"What helped was to consult with each other because you may find that some else understands the problem better than I do, then we would help each other that way."* (**Participant A6**).

*"Collaborated team work will assist in getting information that the lecturer will, accept at the end of the day."* (**Participant Y**).

*"I will advise student to utilize good communication it helps a lot to communicate with others especially when you approaching exams it happens that there are some things that you don't understand and my friend explains to me it is easy to remember what you friend said than remembering what the lecturer said because is intimidating."* (**Participant L**).

Subsequently, other participants mentioned that:

*"What I did I studied was I put more effort and I also ask my fellow class mates to explain some things better to me, because sometimes it's a bit difficult to approach the lectures so I prefer to ask my colleagues to help me where they understand."* (**Participant N**).

*"Studying as a group also helps because discussing amongst ourselves also makes it easier."* (**Participant Q**).

*"To add on what student M was saying I think what also help to overcome the challenges is to get different presentation from different groups as some conditions have similar symptoms so by gathering all that information you will know what makes them different."* (**Participant K**).

3.3.2. Subtheme 1.3.2: Use of Relevant Study Materials Such as Articles, Prescribed Books, and Past Question Papers

The use of relevant study materials such as articles, prescribed books, and past question papers emerged as one of the subthemes. Student nurses reported the use of relevant study materials such as articles, prescribed books, and previous question papers as strategy to overcome the challenges faced during the transition from CBL to PBL. Participants stated that:

*"The right prescribes text books and also relying on online books for more information helped me to improve on my second semester."* (**Participant K**).

*"So firstly, I will start with either google scholar and find information and after I will go to the library for additional information and more sources."* (**Participant W**).

Another participant added that:

*"What helped me was going through past question papers to see how the lecturers are setting- that's my other coping mechanism. So that one helped a lot."* (**Participant Q**).

### 3.3.3. Subtheme 1.3.3: Plan Study Time

The planning of study time also emerged as a subtheme. Student nurses described the proper planning of study time as one of the strategies used to overcome the challenges faced during the transition from CBL to PBL. One of the participants articulated:

*"I did what all the other students did basically. I put more effort and also practice time management. I look on the dates for tests, placement and exams. It is also helps to note them down so that you plan study time for yourself thus give you time to prepare."* (**Participant M**).

Another participant said:

*"Prepare before time like before going to class, so to know the topics that will be done."* (**Participant P**).

### 3.3.4. Subtheme 1.3.4: Consultation with Lecturers

Consultation with lecturers emerged as one of the subthemes. Student nurses reported consultation with lecturers as a strategy to overcome the challenges faced during the transition from CBL to PBL. Participants expressed their views and mentioned that:

*"If I feel like I have questions I usually go and consult with the module facilitator or the relevant person which we are referred to by the module facilitator to consult."* (**Participant U**).

*"Well what I can advise others is that they should consult with the lectures because some of us failed to consult that is where we encounter problems they should consult more and also use the library they can use the scenarios on the question papers."* (**Participant N**).

Another participants added:

*"I also forget to add consultation the lectures give you feedback and show you what to do."* (**Participant P**).

*"The consultations with the lecturers were helpful because they helped us identify the areas that were troubling us too much and we would try to work on them."* (**Participant A5**).

*"Attending class wholeheartedly, make notes and revise after each class in order to identify the areas where challenges are. After that consult with the lecturers with informed information because some lecturers like to ask what is it that you really don't understand and you can't say everything you have to be specific on what you don't understand."* (**Participant M**).

## 4. Discussion

The study explored and described challenges faced by student nurses during the transition from case-based learning to problem-based learning at one of the institutions of higher learning in the North West Province (NWP). The data's findings particularly identified three themes: challenges regarding facilitation, challenges regarding assessment, and strategies to overcome challenges.

### 4.1. Challenges Regarding Facilitation

The transition from case-based learning (CBL) to problem-based learning (PBL) has presented several challenges to student nurses. Challenges regarding facilitation appeared

as the primary theme from the collected data and subthemes were also identified. Many student nurses stressed how challenging it was to transition from one teaching strategy to another. As a result, students worry excessively about their success with PBL as the newly introduced teaching strategy [13]. The same author further reported that student nurses who adapt to the PBL process experience anxiety and fear of the unknown [13]. Taken together, these findings suggest that transitioning from one teaching strategy to another requires the necessary support for student nurses.

The issue of working in groups was also reported to be a serious challenge mainly because of poor participation by most of the group members. In this context, working in groups refers to a teaching–learning method consisting of both collaborative and cooperative learning launched to achieve a common goal [14]. Pahomov (2018) stated that working in groups requires students to not just work with group members, but truly collaborate with peers to respond to a given assignment and reach a common objective [15]. However, this cannot be achieved if there is poor participation from students. Ideally, when all group members participate in group work, this leads to a product that reflects the full integration of participants' diverse skill sets [15].

Despite PBL promoting the notion that students should conduct independent information searches, the difficulty in searching for and finding relevant information on accessible databases was reported by the student nurses, particularly when one is unfamiliar with searching methods. This discrepancy could be attributed to students having difficulty in transitioning to PBL. According to Spry and Mierzwinski-Urban (2018), successful electronic information searches involve a variety of steps, including selecting the right databases for the search, relevant keywords, acceptable headings as key features, and correct spelling [16]. Finding the appropriate information will be challenging if these factors are not considered [16]. However, Scells et al. (2020) holds the view that the process that mostly influences bias is the creation of search strategies [17]. If the findings of [16,17] Spry and Mierzwinski-Urban (2018) and Scells et al. (2020) are accurate, student nurses must keep both findings in mind.

The fact that there is more work given to complete using PBL, and within a short time, was another significant challenge raised by student nurses. Abdelkarim et al. (2018) acknowledge that PBL has a variety of drawbacks, which include but are not limited to, students' preparation which somehow increases their workload and time constrains [8]. In addition, Ghufron and Ermawati's study (2018) established that it is challenging to implement PBL since it requires a lot of time, planning, and work [7]. It may be argued that it is obvious that transitioning from CBL to PBL does in fact raise challenge and time demands on student nurses. Therefore, one possible implication of this is that while using PBL to approach the curriculum, student nurses should be given adequate time.

A further challenge which emerged was lack of proper guidance from lecturers during the transition from CBL to PBL. Bouwmeester et al. (2019) added that well-informed lecturers are essential for fostering critical thinking and guiding student nurses in problem-solving techniques [18]. Salari et al. (2018) further stated that in PBL, lecturers must strive to guide students as their immediate facilitators and must be considered as a coaches who provide guidance to keep students on track [19]. These literature suggests that [18,19] have similar views. In conclusion, in order to transit properly from CBL to PBL, student nurses should be cautiously guided.

Another significant challenge brought by the student nurses was the constant ongoing learning issues with the PBL curriculum. If a group of student nurses cannot agree on a particular aspect of knowledge presented, it is a "learning issue" and homework is assigned to the students. In this context, a "learning issue" is the homework given to studenst. Consequently, the findings of the study suggest that transitioning from CBL to PBL becomes even more challenging because there are more "learning issues", which in turn result in a greater burden for students. In this regard, students are not corrected or provided with the correct answers and they even go to the examination or test without knowing the appropriate knowledge, which is a challenge because it may lead to poor performances.

A study conducted by Songsirisak and Jitpranee (2019) explains the fundamental objectives of homework, which is to evaluate students' understanding and learning progress [20]. Furthermore, it also gives students the chance to enhance their study habits, academic performance, and academic achievements. However, students' perspectives on homework vary depending on their educational backgrounds, worldviews, attitudes, and cultures [20]. Despite the purported objectives and benefits of homework, the study's finding suggests that student nurses are not content with receiving homework and have negative attitudes towards it. As a result, it can be suggested that to balance workload and ensure that students are acquiring enough knowledge at the same time, student nurses should only be assigned a reasonable workload of homework.

The study's findings also revealed that facilitating online problem-based learning was another significant challenge. This means that the student nurses attended their classes online, utilizing a problem-based learning approach. In the study conducted by García-Morales et al. (2021), students who took classes online experienced difficulties with connectivity issues caused by unexpected network outages [21]. Daugvilaite (2021) further added that disruptions to the connection or freezing of the screen during an online lesson caused students to lose focus and interrupted learning [22]. García-Morales et al. (2021) noted additional significant disadvantages of online learning, which include boredom, a feeling of isolation, a lack of time to study various subjects, and a lack of self-organizational skills [21]. The findings of this study support García-Morales's study in a sense that participants in this study also reported the same challenges.

A lack of student instructors was identified by student nurses as another critical challenge. In the context of this study, senior students in their final year who help junior students with a specific module or offer supplementary classes are known as student instructors. The lack of student instructors in third year has caused a major worry among student nurses, who feel that it disadvantages them. Peer facilitation complements adult learning theories as it unifies cognitive, social, and constructivist theories [23]. Additionally, according to Oh et al. (2018), student nurses reported feeling more comfortable sharing their own thoughts in a peer-facilitation situation and the students preferred discussion that is facilitated by their peers [24]. Moreover, students tended to contribute more original ideas and show a more engaged level of participation when peers facilitated dialogues [24]. In view of the above conversation, a similar recommendation to that made by Davis and Richardson (2017), for universities to empower second- and third-year students to peer facilitate learning sessions for first-year students, can be implemented for the third years [23].

*4.2. Challenges Regarding Assessment*

The challenges regarding the assessment of student nurses emerged as a second theme of the study. This entails a lack of feedback and lack of revision. The first subtheme to emerge from this theme was the lack of feedback. Seibert (2021) asserts that lecturers must be aware that student nurses require more encouragement, support, and feedback when participating in PBL for the first time to simplify their transition process [13]. On the other hand, the student nurses expressed their gratitude for the lecturers who provided feedback and support and encouragement [13]. Therefore, students should be motivated, supported, and provided with feedback to reduce any anxiety they may be experiencing as a result of taking part in PBL for the first time, thus simplifying their transitioning process.

The lack of revision by student nurses became the second subtheme for this theme. Students stated that while they struggle to transition entirely to PBL, they finish module content late, which leaves them with less time to revise for tests and examinations causing them to panic. According to Cottrell's study (2017), simply having extra time available for studying and revision can be beneficial in a sense that it somehow allays their anxiety relating to tests and examination [25]. Duret et al. (2018) noted that some students cram in the day or even hours prior to a test or examination [26]. It is clear from the literature and study findings that students require ample time for study and revision, to avoid stress and cramming during

tests and examinations, because they are not really learning the relevant knowledge, skills, and attitudes inherent to the nursing profession.

*4.3. Strategies to Overcome Challenges*

The last theme that emerged from the findings of this study was strategies used by the student nurses to overcome the challenges faced during the transition from CBL to PBL. There were four subthemes that emerged from the main theme. The first subtheme to emerge was peer-assisted learning. According to student nurses, transitioning from CBL to PBL brought many challenges as highlighted in the study's findings. However, students developed certain strategies to overcome those challenges, such as learning from what other students do to deal with those challenges and assisting one another.

The evidence from the study findings suggests that this strategy is corroborated by the study of Abdullah and Chan (2018), which states that learning with and from peers enables one to gain learning outcomes, including teamwork, critical thinking, communication, and other skills such as coping mechanisms [27]. Similar to this, Johnson and Johnson's study (2018) established that peer-assisted learning involves classmates of equal status actively helping one another with learning-related issues [28]. Additionally, peer-assisted learning is built on collaboration because support and motivation rarely include competitive engagement [28].

Another of the subthemes that emerged was the usage of pertinent study materials, such as peer-reviewed articles, prescribed books, and past question papers. The utilization of such materials was reported by student nurses as one of the strategies to the overcome challenges they encountered during the transition from CBL to PBL. Most institutions and individuals share their learning resources on the Internet in an open and cost-free manner as open educational resources (OER), even though they are frequently regarded as important intellectual property in the competitive higher education industry [29]. According to Hylén (2021), OER are broadly characterized as digital materials that are freely and openly made available for teachers, students, and self-learners to use and re-use for teaching, learning, and research [29].

Within the higher education context, Colvard et al., (2018) state that OER normally include free, online learning content, software tools, and accumulated digital curricula that are not restricted by copyright licenses and are available to retain, reuse, revise, remix, and redistribute (5Rs). From the literature, it is clear that the usage of pertinent study material is beneficial to the students [30].

Planning study time emerged as another crucial subtheme. One of the strategies used by student nurses to overcome challenges experienced during the transition from CBL to PBL was proper time management. However, Adams and Blair (2019) state that many students struggle to strike a balance between their daily activities and their studies [31]. Effective time management is linked to better academic achievement and decreased levels of anxiety in students, as students learn coping strategies that allow them to negotiate competing demands [31,32]. Additionally, students today frequently complain that they do not have enough time to finish all the duties that have been given to them [32]. The literature supports the concept that suggest that students who plan their study time prosper academically.

Another important subtheme that emerged from the main theme was consultation with lecturers. Student nurses mentioned consultation with lecturers as a strategy for overcoming challenges during the transition from CBL to PBL. According to Agustin et al. (2020), the challenge could be students who are failing academically, which can affect other aspects of the students' lives [33]. One of the strategies to address this is to consult with lecturers who can provide counselling and guidance services, with a primary focus on academic and personal social guidance for students [33].

Furthermore, the knowledgeable lecturers can advise students to direct their explorations to accelerate learning [34]. However, the recommendations of other studies suggest that lecturer must be an expert in the subject being learned to advise students accord-

ingly [34]. Therefore, the literature supports the student nurses' route of consulting with lecturers to assist them with the challenge they experience during the transition period.

## 5. Conclusions

The study explored and described challenges faced by student nurses during the transition from case-based learning to problem-based learning at one higher institution of learning in the North West Province of South Africa. The study has identified that student nurses were faced with different challenges during the transition from CBL to PBL teaching strategies. Therefore, necessary support should be given to students in order for them to cope with these challenges. However, student nurses suggested strategies that could also be used to overcome such challenges. Furthermore, the findings reported here provided new light that student nurses who are accustomed to CBL may resist taking more responsibility for their own learning when they transition to PBL. This might happen even though there is evidence in the literature which indicates that PBL adds value to the educational experience.

**Author Contributions:** This manuscript is based on RJP's partial fulfilment of the requirements for a Master of Nursing Science (MNSc) in Community Nursing under the supervision of B.J.M. and M.J.M. at the NWU. The draft of the manuscript was written by R.J.P. and its finalization was equally completed by B.J.M. and M.J.M. All authors have read and agreed to the published version of the manuscript.

**Funding:** The author(s) received no financial support for the research, however the publication fee of this article will be paid by the North West University (NWU).

**Institutional Review Board Statement:** The study was conducted in accordance with the Declaration of Helsinki and approved by Health Research Ethics Committee (HREC) of the North West University (Ethics reference number: NWU-00218-21-s1 on 21 January 2022).

**Informed Consent Statement:** Written informed consent was obtained from all participants involved in the study.

**Data Availability Statement:** The authors confirm that the data are available. However, it cannot be shared with anyone as per agreement established with participants according to research regulations and Protection of Personal Information Act (POPIA), protected by HREC. However, the data that support the findings of this study are available within the article.

**Acknowledgments:** We acknowledge the authors of sources cited in this study.

**Conflicts of Interest:** The authors declare no conflict of interest and have no financial nor personal relationship that could have unduly influenced the writing of this article.

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
