# Peer review of "Challenges Regarding Transition from Case-Based Learning to Problem-Based Learning: A Qualitative Study with Student Nurses"

_nursrep, doi:10.3390/nursrep13010036_

Round 1
Reviewer 1 Report
Dear authors,
congratulations for your work. The topic is interesting and will be good for international readers. Results of plagiarism analysis were 9.2% which is great. However there many things your should improve.
Title - this title is too long, and should have research design. I recommend to shorten title and add your research design.
Abstract is well written
Introduction - please try to better explain what is scientific contribution of your research work, whats added value.
Methodology - please add ethical concerns, which body approved the research, what about informed consent, have you followed Helsinki declaration. It would be nice to cite Helsinki declaration.
Results - Your result section is too short and inadequate. Please present your work in better tables, citations.
Reviewer 2 Report
Attached the comments

Round 2
Reviewer 1 Report
Dear authors,
thanks for your replies and for sending improved manuscript. From my prospective I am satisfy with provided corrections and don't have anything else to add.